# Systematic Analysis of the *CCoAOMT* Gene Family in *Isatis indigotica* and the Molecular Mechanism of *CCoAOMT8*-Mediated Flavonoid Synthesis Under Alkaline Stress Treatment

**DOI:** 10.3390/biology14111518

**Published:** 2025-10-30

**Authors:** Bo Liu, Lingyang Kong, Junbai Ma, Shan Jiang, Lengleng Ma, Jiao Xu, Weichao Ren, Wei Ma

**Affiliations:** 1College of Pharmacy, Heilongjiang University of Chinese Medicine, Harbin 150040, China; liubo870617@163.com (B.L.);; 2College of Jiamusi, Heilongjiang University of Chinese Medicine, Jiamusi 154000, China

**Keywords:** *Isatis indigotica*, CCoAOMT gene, evolutionary analysis, qRT-PCR, yeast monoculture, chemical composition

## Abstract

**Simple Summary:**

This study aims to identify and analyze the *CCoAOMT* gene family in *Isatis indigotica* at the whole-genome level and predict its impact on the growth and development of *Isatis indigotica*. Simultaneously, this study investigates the expression characteristics and response patterns of the *CCoAOMT* gene in *I. indigotica* under abiotic stress treatment and determines the content of flavonoids in *Isatis indigotica* roots. In addition, yeast one-hybrid experiments confirmed that *IiWRKY48* and *IiWRKY54* can activate the promoter of the *CCoAOMT* gene in *I. indigotica*, providing a key basis for further understanding the role of this gene in physiological processes and the secondary metabolism regulation of *I. indigotica*.

**Abstract:**

Caffeoyl CoA O-methyltransferase (*CCoAOMT*) is one of the key regulatory enzymes in the lignin biosynthesis pathway. In addition, it participates in the modification of flavonoids, which significantly impacts plant growth, development, and antioxidant capacity, and it plays a crucial role in plant responses to stress and adversity. There is a current research gap in the *CCoAOMT* gene family of *Isatis indigotica*, particularly in terms of systematic identification and functional validation. Therefore, this study employed bioinformatics techniques to determine the composition of the *CCoAOMT* gene family in the *Isatis indigotica* genome. Eight members of the *IiCCoAOMT* gene family were identified, and their gene structures and motifs are relatively conserved. These members of the *IiCCoAOMT* family are located on three different chromosomes (3, 6, and 7) and exhibit significant tandem replication. According to phylogenetic research, *IiCCoAOMT* is divided into four distinct groups: Ia, Ib, Ic, and II. It is worth noting that the *IiCCoAOMT* genes in group Ia may be candidate genes involved in flavonoid biosynthesis and indirectly affect the content of flavonoid components., Subsequently, a yeast one-hybrid experiment verified that *IiWRKY48* and *IiWRKY54* could activate the *CCoAOMT* gene promoter in *Isatis indigotica*. These results provide a theoretical basis for understanding the function of *CCoAOMT* genes.

## 1. Introduction

*Isatis indigotica* is a biennial herbaceous plant of the Brassicaceae family [1]. It is a popular Chinese herbal medicine, and its medicinal parts are roots and leaves. The leaves of indigo, commonly known as Folium Isatidis [2], are widely used in the treatment of various diseases [3]. In traditional Chinese medicine, the basic theory attributes functions such as clearing heat and detoxifying, cooling the blood, and removing macules. The main components of indigo are lignin, indole alkaloids, and polysaccharide components [4]. These components have good clinical therapeutic effects in preventing and treating viral infections, immunodeficiency diseases, and cancers, such as influenza, mumps, pneumonia, allergic dermatitis, acute respiratory syndrome, and other diseases [5]. Plants face environmental fluctuations throughout their life cycles, including abiotic and biotic stress treatments. Therefore, plants develop various response mechanisms to adapt to these unfavorable conditions, and many genes are activated for expression. The expression level of stress treatment response genes may contribute to the survival of plants [6]. Typically, the control of gene expression in response to external elements is predominantly governed by families of enzymes, which are crucial adaptation mechanisms in plants [7].

The external environment has a significant influence on the normal growth and development of plants. External stress treatments such as drought, low temperatures, and saline–alkali stress treatment will cause different degrees of damage in plants [8], which may even lead to plant death [9]. Secondary metabolites in plants play a vital role in stress treatment reactions and are essential for their growth and development. The pathways of phenylpropane metabolism include lignin and flavonoids, which have different effects on stress treatment resistance. Lignin is one of the main substances that make up the plant’s cell wall. It can increase the hardness of the cell wall, play a better supporting role, and exhibit strong compressive resistance while also providing functions in disease resistance and stress treatment resistance [5]. Research has shown that high-nitrogen applications can lead to lignin deposition and reduced content [10]. As potent reactive oxygen species scavengers, plant flavonoids exhibit a strong antioxidant effect, responding effectively to various stress treatments and improving the tolerance and resistance of plants to both biotic and abiotic stress treatments. The latest research indicates that *CCoAOMT* is involved in lignin biosynthesis, as well as in erucic acid metabolism and isorhamnetin biosynthesis, through its role in methylating hydroxycinnamic acid [11] or flavonoid precursors [12]. The content of polymethoxyflavones in citrus fruits is significantly increased after being infected with Huanglongbing [13]. In addition, flavonoids play a crucial role in the growth and development of plants, as well as in the quality formation process of flowers and fruits. For example, flavonoid compounds are the main substances that affect flower color. They can induce the colonization of rhizobia and arbuscular mycorrhizal fungi in plant roots and improve the absorption of elements such as nitrogen and phosphorus by plants [14].

The year 1988 marked the discovery of *CCoAOMT* in carrot and parsley tissue culture cells, which was linked to their defensive mechanisms against fungal infections [15]. Caffeoyl-CoA O-methyltransferase (*CCoAOMT*) participates in the modification process of O-methylation. Unmodified flavonoids have poor lipid solubility and water solubility. Post-modification effects such as methylation, glycosylation, and hydroxylation can effectively improve the various biological activities of flavonoids and enrich the structure and function of flavonoids [8]. *CCoAOMT* participates in the methylation process of flavonoids [16]. *CCoAOMT* is dependent on Mg^2+^ and includes true *CCoAOMT* that specifically catalyzes caffeoyl-CoA and *CCoAOMT*, which have a preference for catalyzing caffeic acid esters and flavonoids [17]. For example, under drought stress treatment conditions, the expression level of the *CCoAOMT* gene in *Paeonia ostii* leaves increased significantly. When the *PoCCoAOMT* gene of *Paeonia ostii* was transferred into tobacco and subjected to drought treatment, it was found that tobacco plants with the *PoCCoAOMT* gene had stronger drought resistance than wild-type plants. [18]. In *Arabidopsis*, *CCoAOMT1* plays a role in resisting drought stress treatment through the control of H_2_O_2_ buildup and the ABA and ROD signaling routes [19]. Eliminating the *CCoAOMT* enzyme gene in Arabidopsis led to a reduction in G lignin levels and a rise in S and H lignin amounts [20,21]. Additionally, the elimination of this gene results in an excessively sensitive state to salt stress treatment due to the prevention of root growth [22]. *Tobacco* plants with the caffeoyl-CoA-O-methyltransferase gene of the M14 line of sugar beet have strong resistance to salt stress treatment and drought stress treatment [23]. Throughout the growth and maturation of Lonicera macranthoides, the expression intensity of the *LmCCoAOMT* gene correlates with chlorogenic acid levels and plays a crucial role in chlorogenic acid synthesis [24]. The *CCoAOMT* gene also has a certain impact on resistance to bacterial wilt. The expression level of the *CCoAOMT* gene is higher in tomato varieties resistant to bacterial wilt and lower in susceptible varieties [25]. Genomic studies reveal the presence of *CCoAOMT* genes in Arabidopsis thaliana [20], rice [26], and poplar [27] (in the presented order). The *CCoAOMT* gene consists of eight preserved motifs labeled A to H [28].

High-quality genomic data on *I. indigotica* have been published [29]. Nonetheless, research into the evolutionary link and functional confirmation of the *CCoAOMT* family in *I. indigotica* remains scarce. This article determines the *CCoAOMT* gene family in the genome of *I. indigotica* through bioinformatics analysis, thus filling this gap. The qRT-PCR (quantitative real-time polymerase chain reaction) technique is used to detect gene expression in this family of leaves under alkali stress treatment at different time periods and to determine the content of chemical flavonoid components. The findings aid in comprehending *CCoAOMT* genes and offer fresh insights for additional studies on their traits.

## 2. Materials and Methods

### 2.1. Experimental Materials

The seeds used in the experiment were all from the Heilongjiang University of Traditional Chinese Medicine. Firstly, treat the samples with 70% (*v*/*v*) ethanol for 1 min; then, soak and disinfect the sample with a solution containing 0.5% (*v*/*v*) sodium hypochlorite for 10 min, and finally, rinse the sample 3–5 times with sterile distilled water. During the subsequent 24 h pretreatment period, the seeds were placed in a light-controlled incubator under strict growth conditions: temperature of 22 °C, relative humidity of 65%, light cycle of 16 h/8 h in the dark, and light intensity of 120 μ mol·m^−2^s^−1^. The seeds were sown in a flower pot made of vermiculite and nutrient soil at a ratio of 1:4 until 3–4 leaves grew, and they were transplanted into a pot with a diameter of 12 cm and further cultured for about 60 days. Select materials with good growth conditions and stable growth, and then, place them in a light-controlled incubator for 24 h to remove any damage. The initial pH value of the mixed nutrient soil was measured before the experiment (using a pH meter with a soil water ratio of 1:5) [30]. The measurement results showed that its pH value was 6.0 ± 0.2, which met the growth requirements of the experimental species. During the experiment, a 200 mmol NaHCO_3_ alkaline solution was poured every 24 h, with a pH value of 8.2 [31]. Meanwhile, the control group received an equal amount of sterile water for irrigation on the same day of treatment. Samples were taken at 0, 1, 3, and 5 days under the same cultivation and management conditions. During RT-PCR, flavonoid content, similarity evaluation, POD activity, and MDA content experiments, we selected three independent samples as biological replicates and set three technical replicates for each independent sample to ensure the accuracy and stability of the experiment. The newly collected samples were rapidly frozen using liquid nitrogen and stored in a refrigerator set to −80 °C.

### 2.2. Identification and Characterization of IiCCoAOMT Genes

We downloaded the *Isatis indigotica* genome file and referenced the *isatis indigotica* genome annotation information file from FIGHARE (https://figshare.com/, accessed on 17 July 2025). Then, the protein sequences of the *CCoAOMT* of *Arabidopsis thaliana* L. were downloaded from TAIR (https://www.arabidopsis.org/, accessed on 17 July 2025) [32] as reference sequences, and according to the SMART website (http://smart.embl-heidelberg.de/, accessed on 17 July 2025), the sequences obtained in the previous step were subjected to structure prediction. Sequences without the typical domain of the CCoAOMT protein were removed. Sequences with missing or incomplete domains were manually removed. When dealing with several transcripts of a single gene, the lengthiest transcript was chosen as the standard sequence. The remaining protein sequences were regarded as members of the *IiCCoAOMT* family. Using ExPASy (http://web.expasy.org/protparam/, accessed on 18 July 2025), the website predicts the physicochemical properties of the *Isatis indigotica* IiCCoAOMT protein [33]. Predicting the subcellular positioning of the IiCCoAOMT protein is possible via the Plant-mPLoc website (http://www.csbio.sjtu.edu.cn/bioinf/plant-multi/, accessed on 18 July 2025) [34]. The tertiary structure of the protein encoded by *I. indigotica CCoAOMT* is predicted, and protein homology tertiary structure modeling is performed using SWISS-MODEL (https://swissmodel.expasy.org/, accessed on 18 July 2025).

### 2.3. Analysis of the Gene Structure of I. indigotica CCoAOMT and the Conserved Motifs

Using the GFF annotation file of the *Isatis indigotica* genome, the intron–exon distribution of the *Isatis indigotica CCoAOMT* gene was obtained. Protein sequence alignment analysis was performed through the online software MEME Suite v5.5.2 (https://meme-suite.org/meme/, accessed on 18 July 2025) [35]. Set the significance threshold for E value to <1 × 10^−5^. The conserved motifs identified in the Isatis indigotica CCoAOMT protein were identified. The refined settings include the distribution of sites, assigning either 0 or 1 to each sequence; a motif count of 8; and preset values for additional parameters. TBtools (v2.136) software was used for multiple sequence alignment.

### 2.4. Phylogenetic Analysis of the Isatis indigotica CCoAOMT Gene

We performed multiple alignments of the selected *CCoAOMT* amino acid sequence using Clustal X v2.0 [36], and the phylogenetic tree was constructed by the neighbor-joining (NJ) method using MEGA11 software [37] (https://megasoftware.net, accessed on 18 July 2025) with 1000 bootstrap replicates. Classify the members of the *CCoAOMT* gene family in *I. indigotica* based on the classification method and phylogenetic relationships of the *CCoAOMT* gene family.

### 2.5. Analysis of Gene Cis-Acting Elements

Extract the 2000 bp upstream sequence of the transcription start codon for all members of the *CCoAOMT* gene family and submit it to the online website PlantCARE [38] (https://bioinformatics.psb.ugent.be/webtools/plantcare/html/, accessed on 18 July 2025). The website was used for predicting cis-acting components, and a self-written R-3.6.0 script was used for statistical analysis and the visualization of the prediction results.

### 2.6. Gene Chromosome Location and Collinearity Analysis

Obtain the gene structure annotations of the *CCoAOMT* gene from the database and retrieve chromosome positions from these annotations. TB-Tools 2.142 is used to display the accurate location of genes on each chromosome. Collinearity analysis was conducted on its genome using TBtools (v2.136), and it was analyzed with the genomes of *Arabidopsis thaliana* and grape, with visual analysis conducted.

### 2.7. RNA Extraction from Leaves at Different Treatment Times and Real-Time Fluorescence Quantitative PCR Analysis

The leaves of *Isatis indigotica* were washed clean with tap water.The SPAREasy Plant RNA Kit (AC0305) was used to extract total RNA from *Isatis indigotica* plants (Shandong Sparkjade Biotechnology Co., Ltd., Jinan, China). Subsequently, RNA quality was assessed using 1% agarose gel electrophoresis, followed by its reverse transcription into cDNA. Utilize a nanodrop spectrophotometer to ascertain the concentration of the acquired cDNA. Finally, use the extra kit to perform qRT-PCR experiments with cDNA as a template. The PCR procedure is as follows: 94 °C for 30 s; 45 cycles of 94 °C for 12 s; 58 °C for 30 s; and 72 °C for 45 s. Then, heat the sample at 79 °C for 1 s to read the plate. After the end of the last PCR cycle, increase the temperature from 55 °C to 99 °C at a rate of 0.5 °C/s to generate the melting curve of the sample. Use the EF1α gene as an internal reference gene [39]. Three biological replicates and three technical replicates were performed for each sample. The 2^−ΔΔCt^ algorithm was employed to determine the comparative expression levels of each gene. Set three technical replicates for each experiment [40]. The above test steps are carried out according to the kit’s instructions. Use Primer (version 5.0) software to design qPCR primers. Beijing Ruiboxingke Biotechnology Co., Ltd. (Beijing, China) is the sole producer of all primers employed in this research.

### 2.8. Determination of Flavonoid Chemical Constituents in I. indigotica Tissue Samples

The experimental materials are *Isatis indigotica* leaves used at different time periods under the alkali stress treatment. The standard products are purchased from Sichuan Weikeqi Biological Technology Co., Ltd. (Chengdu, China), including luteolin (batch number WRQ-0000398), isoquercitrin (batch number DSTDY000603), and Folium Isatidis control herb (ycwp24062805). Methanol is of chromatographic grade, and the other reagents are of analytical grade. Water is ultrapure water. The following were used: Waters (Milford, MA, USA) XTERRA^®^ MS C18 (250 mm × 4.6 mm, 5 μm); mobile phase: methanol (A) 0.1% formic acid aqueous solution (B); gradient elution: 0–75 min, 10% → 64% A; 75–80 min, 64% → 90% A; flow rate: 1.0 mL/min; detection wavelength: 254 nm; temperature: 35 °C; injection volume: 10 μL [41]. Three biological replicates and three technical replicates were analyzed for each sample.

Creating a Standard Solution: To craft a mixed reference solution, take an appropriate amount of luteolin-7-O-glucoside and isoquercitrin reference substances, weigh them precisely, dissolve them in a 70% methanol solution, and quantitatively dilute them to prepare a mixed reference solution containing the above reference substances at the desired concentration per 1 mL.

Preparation of the Test Solution: Weigh 0.5 g of Folium Isatidis medicinal material; cut it into pieces; place it in a stoppered conical flask; accurately add 10 mL of 70% methanol solution; seal it tightly, weigh it, and extract it by ultrasound (frequency: 50 kHz) for 45 min; weigh it again; make up the lost weight with 70% methanol solution; shake it well; filter it with a 0.45 μm organic microporous membrane; and take the subsequent filtrate to obtain it. Compare the fingerprint spectra of *Isatis indigotica* treated with alkali at different time periods, determine them according to the above chromatographic method, and evaluate the similarity using *Isatis indigotica* as the control medicinal material.

### 2.9. Yeast One-Hybrid Assay

Under abiotic stress treatment, *WRKY* transcription factors can specifically bind to the promoter W-box (TTGACC) element. The screening of W-box elements that bind to the *IiCCoAOMT* promoter and *WRKY* transcription factors was conducted through the plantCARE website. By reviewing the literature and constructing the *WRKY* gene family of *Isatis indigotica*, it was determined that *WRKY48* and *WRKY54* have regulatory co-functions. Therefore, yeast one-hybrid analysis was used to analyze the interactions between *WRKY48* and *54* and the promoters of the *IiCCoAOMMT3*, *IiC-CoAOMT4*, and *IiCCoAOMMT7* genes [42]. The promoters for the quartet of *CCoAOMT* enzyme genes were inserted into the pLacZi vector, using EcoR I and Sac I sites as baits. Cloning of the wrky transcription factor’s CDS region into the pGADT7 vector occurred using the EcoR I site as the target. The plasmids pLacZi-IiCCoAOMT3, pLacZi-IiCCoAOMT4, and pLacZi-IiCCoAOMT7, along with the blank vector pLacZi, were introduced into the EGY48 strain and grown in media with varying concentration levels to identify toxicity and self-activation. The vectors of *WRK48*, *54*, and the recombinant vectors pLacZi-IiCCoAOMT3, pLacZi-IiCCoAOMT4, and pLacZi-IiCCoAOMT7 were transformed into EGY48 yeast. The recombinant plasmids of pLacZi and JG4-5, pLacZi and IiWRKY54-JG4-5, and JG4-5 and pLacZi-IiCCoAOMT7 were co-transformed into yeast strain EGY48 as negative controls.

The transformed yeast was dispersed on an SD/−Trp-Ura-Broth medium and cultured at 30 °C for 3 to 5 days. Positive colonies were picked and cultured in SD/−Trp-Ura-Broth liquid medium and then cultured on chromogenic medium at 30 °C for 1 to 3 days. Based on color development, it can be determined whether the reporter gene is activated. Three biological replicates and three technical replicates were performed for each sample.

### 2.10. Determination of Antioxidant Content in I. indigotica 

The malondialdehyde content assay kit(BN0049-W96) and POD (BN0048-W96) enzyme activity assay kit (Shandong Sparkjade Biotechnology Co., Ltd., Jinan, China) were used to determine the malondialdehyde content in the tissue samples of *Isatis indigotica*. Weigh approximately 0.1 g of tissue (or 0.5 g for adequately moist samples), introduce 1 mL of the extraction mixture, blend in an ice bath, centrifuge at 4 °C × 12,000 rpm for 10 min, collect the supernatant, and store it on ice for analysis. Combine the active solution with the sample as needed, transfer 200 μL of the supernatant into a 96-well plate, and measure absorbance A at wavelengths of 532 nm and 600 nm, ΔA = A532-A600. Calculate the malondialdehyde content using the following formula: MDA (malondialdehyde) content (nmol/g fresh weight) = 32.3 × ΔA ÷ W, where W is the sample mass, g. Weigh about 0.1 g of tissue (0.25 g for samples with sufficient moisture), add 1 mL of extraction solution, and perform ice bath homogenization. Place the supernatant in a centrifuge at 4 °C and 12,000 rpm for 10 min; then, chill it on ice for analysis. Mix the test solutions, reagent 1, reagent 2, and reagent 3, in a 96-well plate as required. Immediately read the absorbance value A1 at 470 nm, and after 1 min, read A2, ΔA A2-A1. The following was used: POD (Δ OD470/min/g fresh weight) = 100 × ΔA ÷ W, where W denotes sample mass, g. POD (peroxidase) enzyme activity is defined as increasing the absorbance value at 470 nm by 1 unit U per gram of tissue per minute in the reaction system. The above experiment involves three biological replicates and three technical replicates for each sample.

### 2.11. Statistical Analysis

The biological experiment was conducted with three biological and technical replicates, and statistical evaluation was performed using one-way analysis of variance (Tukey test). Measurement information is presented in the form of the mean plus or minus the standard deviation. Use *p* < 0.05 as the benchmark for statistical correlation and *p* < 0.01 as the benchmark for significance comparison.

## 3. Results

### 3.1. Identification and Physicochemical Property Analysis

Eight predicted *IiCCoAOMT* genes were identified from the *Isatis indigotica* genome (Appendix A). The *IiCCoAOMT* genes contain two conserved domains: UDP-glucuronosyl and UDP (Glucosyl transferase and 0-methyltransferase (PF01596.17)) [43]. Some genes also have only one domain: O-methyltransferase. Furthermore, an analysis was conducted on the dimensions of the protein, the isoelectric point, its molecular mass, and the subcellular positioning of *IiCCoAOMT* (Table 1). The findings revealed that the briefest segment, the IiCCoAOMT5 protein, comprises 147 amino acid units. The number of amino acid residues in the remaining seven sequences is around 200. The lengths of the *IiCCoAOMT* gene sequences vary greatly. Among them, the *IiCCoAOMT2* gene sequence is the longest (834), and the gene sequence lengths are between 400 and 850 bp. The theoretical isoelectric points of proteins range between 4.96 (*IiCCoAOMT3*) and 9.37 (*IiCCoAOMT5*). It has been determined that there are both acidic and alkaline proteins. The prediction results of the subcellular localization show that *IiCCoAOMT2*, *IiCCoAOMT3*, *IiCCoAOMT4*, and *IiCCoAOMT6* are located in the chloroplast, *IiCCoAOMT5* is located in the nucleus, *IiCCoAOMT7* is located in the peroxisome, and *IiCCoAOMT8* is situated in the cytoplasm. The basic details of all members of the *IiCCoAOMT* family are summarized in Table 1. Using the SWISS-MODEL tool for the homology modeling of *IiCCoAOMT*, a three-dimensional model is constructed (Figure 1). It can be seen that *IiCCoAOMT* is composed of two subunits, contains abundant α-helices and random coils, and has two SAM binding regions. As shown in the figure, the three-dimensional model of *IiCCoAOMT2* differs significantly from that of other *IiCCoAOMT* genes. A genetic mutation has likely occurred.

### 3.2. Analysis of the Gene Structure of I. indigotica CCoAOMT and the Conserved Motifs and Sequence Alignment of Its Encoded Protein

To further study the similarity and diversity of motifs in different IiCCoAOMT proteins, MEME software is used for protein domain prediction (Figure 2 and Appendix A). The results show that the protein structure of *IiCCoAOMT* is relatively conserved. A total of eight relatively conserved motifs (named Motif1~8, respectively) were identified. Among them, Motif1, Motif6, and Motif7 are the most conserved and are common motifs in each group (Figure 2A). In addition to the common motifs, each group of motifs also has certain specificities. Motif8 is unique to the *IiCCoAOMT7* and *IiCCoAOMT8* genes. Such motifs may be related to the diversity of functions of the *IiCCoAOMT* gene. The variation between exons and introns plays a crucial role in the evolution of gene families, offering compelling proof for analyzing evolutionary paths [44,45]. For a better understanding of *IiCCoAOMT’s* gene architecture, we acquired the genomic and CDS sequence details of each gene based on *Isatis indigotica’s* genomic data, followed by the creation of an exon–intron structure diagram. (Figure 2B). The results show that the number of exons and introns per *IiCCOAOMT* family member was relatively conserved. Most *IiCCoAOMT* genes contain five exons and four introns. This indicates that four introns are the main structural form of the *IiCCoAOMT* gene. Multiple sequence alignment was performed on eight IiCCoAOMT proteins to study their conserved protein domains with respect to existence and location (Appendix A). The *IiCCoAOMT* gene contains two conserved domains: UDP-glucuronosyl and UDP (Glucosyl transferase and 0-methyltransferase (PF01596.17)). Some genes also have only one domain, O-methyltransferase, which is similar to the results of other plant species.

### 3.3. Phylogenetic Analysis of IiCCoAOMT Family Members

To determine the evolutionary relationship between *IiCCoAOMT* and *CCoAOMT* of other model plants, a phylogenetic tree was constructed using 37 *CCoAOMTs* from four plant species (17 from *Dendrocalamus farinosus* [15], 7 from *Glycine max*, 6 from rice, and 7 from *Arabidopsis*) (Appendix A). The results show that the eight *IiCCoAOMT* genes are divided into four subfamilies (Figure 3). Clustered CCoAOMT protein sequences within a single branch tend to exhibit more functional resemblance. The *I. indigotica CCoAOMT* family proteins are divided into three subgroups: Subgroup Ia contains two IiCCoAOMT proteins, namely, *IiCCoAOMT3* and *IiCCoAOMT4*. Subgroup Ib contains four IiCCoAOMT proteins, namely, *IiCCoAOMT1*, *IiCCoAOMT5*, *IiCCoAOMT6*, and *IiCCoAOMT7*. Subgroup II contains two IiCCoAOMT proteins, namely, IiCCoAOMT2 and IiCCoAOMT8. The majority of members grouped under the same subgroup exhibit comparable motif configurations, suggesting that proteins derived from the same sub-branch might serve similar roles. Research revealed that both *I. indigotica* and the prototype plant Arabidopsis are members of the Brassicaceae family. In each subfamily, the *IiCCoAOMT* gene is closest to the *AtCCoAOMT* gene. For example, these include *AtCCoAOMT3* and *IiCCoAOMT8* and *IiCCoAOMT4* and *AtCCoAOMT1*, indicating that the evolutionary relationship between the members of the *IiCCoAOMT* family and the members of the *AtCCoAOMT* family is the closest. Consequently, the gene functions might be analogous between the *IiCCoAOMT* gene family members and those in the *AtCCoAOMT* gene family.

### 3.4. Cis-Acting Element Analysis of IiCCoAOMT Gene Promoter

To explore the mechanism of action of the *IiCCoAOMT* gene in stress treatment response and development, the PlantCARE online tool was used to analyze the 2000 bp upstream promoter sequence of the *IiCCoAOMT* gene and its cis-acting elements (Figure 4). A total of 42 core promoter elements were identified and classified into six categories according to their functions: plant hormone response, light response element, abiotic stress treatment response, plant growth hormone, and core. A great number of cis-acting elements were found upstream of the eight *IiCCoAOMT* genes, revealing that *light signals regulate IiCCoAOMT*. Furthermore, a multitude of plant hormone reaction components exist, including ABRE and TGACG-Motif, alongside binding locations for transcription factors like *MYB* and W-box and stress treatment response elements such as STRE, WUN motif, and LTR. In addition, the W-box is exclusive to *IiCCoAOMT1*, *IiCCoAOMT3*, *IiCCoAOMT6*, and *IiCCoAOMT7*, suggesting that the *WRKY* transcription factor may specifically bind to these four members of the *IiCCoAOMT* gene family. *IiCCoAOMT3*, *IiCCoAOMT4*, *IiCCoAOMT5*, *IiCCoAOMT6*, and *IiCCoAOMT7* contain methyl jasmonate (MeJA) response elements (CGTCA motif and TGACG motif). Among them, *IiCCoAOMT4* contains the most MeJA response elements. These findings suggest that multiple hormones control *IiCCOAOMT*. It is worth noting that the upstream regions of the *IiCCoAOMT1*, *IiCCoAOMT2*, *IiCCoAOMT3*, *IiCCoAOMT4*, *IiCCoAOMT5*, *IiCCoAOMT6*, and *IiCCoAOMT7* genes all contain MYB binding sites involved in the regulation of flavonoid biosynthesis.

### 3.5. Chromosome Localization and Homology Analysis of IiCCoAOMT Genes

A gene’s chromosomal position is shaped by its evolutionary history. Therefore, our study shows that the members of the *IiCCoAOMT* gene family are randomly distributed on three chromosomes (Figure 5). According to the chromosomal distribution of *CCoAOMT*, it is named *IiCCoAOMT1-IiCCoAOMT8*. Among them, there are four genes on chromosome 7, and there are two genes on chromosomes 3 and 6 (Figure 5B). From this, it can be seen that the *CCoAOMT* gene is irregularly distributed on the chromosome [46,47]. To elucidate the evolutionary link among *IiCCoAOMT* genes, we examined their interspecific collinearity. However, through analysis, we only found one homologous pair (*IiCCoAOMT1*, *IiCCoAOMT5*, *IiCCoAOMT6*, and *IiCCoAOMT7*) (Figure 5A). The duplicate gene pairs in *IiCCoAOMT* may have similar functions, indicating that these chromosomal fragments may not be differentiated entirely during evolution and may have redundancy in function [48]. To further understand the evolutionary relationship of *IiCCoAOMT*, we constructed an interspecific collinearity analysis diagram of three species: *I. indigotica*, grape, and *Arabidopsis*. According to the analysis, there are only four collinear relationships between *I. indigotica* and *I. indigotica* alone, and there are a total of four collinear relationships among the three species of *I. indigotica*, grape, and *Arabidopsis* (Figure 5C). From the above analysis results, the homology between *I. indigotica* and Arabidopsis is slightly higher.

### 3.6. Analysis of IiCCoAOMT Gene Expression Patterns Under Different Time Periods of Alkali Treatment

The cultivation of *I. indigotica* on a large scale in Northeast China is significantly influenced by its ability to withstand cold and its salinity–alkalinity balance. It is reported that cold tolerance and salinity–alkalinity can affect the content of flavonoids and, at the same time, improve the stress treatment recovery ability of *I. indigotica*. The expression level of genes under alkali treatments at different time points in plants can reflect their functions and possible regulatory pathways to a certain extent, and this is closely related to the growth and development of plants. This study found that when *I. indigotica* was treated with alkali for 0d, 1d, 3d, and 5d, the leaves of *I. indigotica* gradually wilted as time went by, while the normal control group did not show leaf wilting, forming a sharp contrast with the treatment group (Figure 6).

To further verify the function of the *IiCCoAOMT* gene and analyze the impact of alkali stress treatment on the *IiCCoAOMT* gene, through qRT-PCR technology, we verified that as the treatment time prolongs, the change range of the *IiCCoAOMT1* gene is relatively small, and the overall state is gradually decreasing. The genes *IiCCoAOMT2*, *IiCCoAOMT5*, and *IiCCoAOMT8* first increased and then decreased; the genes *IiCCoAOMT3* and *IiCCoAOMT4* first increased and then decreased; the genes *IiCCoAOMT6* and *IiCCoAOMT7* showed a gradually increasing trend (Appendix A). In summary, the qRT-PCR verification results reveal that the *IiCCoAOMT* genes show different expression patterns in response to alkali stress treatment, indicating that these genes are specific in expression (Figure 7).

### 3.7. Analysis of Flavonoid Metabolite Content of IiCCoAOMT Gene

The content of luteolin, isoquercetin, and flavonoids in *I. indigotica* treated with alkali stress treatment at different time periods was detected. The results showed that the content of isoquercetin and luteolin changed at different treatment times. The content of quercetin first increased and then decreased, showing an overall downward trend. It reached its peak on the first day, which was twice the content of the blank group. The range of content variation of luteolin is relatively small, with almost no change in content, showing a first decreasing and then increasing trend, with a subsequent decrease (Figure 8). Comparing Figure 6 with Figure 7 reveals that the content change in isoquercetin is consistent with the expression trend of the *IiCCoAOMT8* gene. The *IiCCoAOMT8* gene may regulate the expression of the flavonoid metabolite isoquercetin. Nonetheless, the variation in luteolin’s content aligns with the expression pattern of *IiCCoAOMT1*. The *IiCCoAOMT1* gene may directly or indirectly regulate the expression of luteolin.

The similarity evaluation software for traditional Chinese medicine chromatographic fingerprint analysis was used in our analysis (Figure 9). The characteristic response curve and fingerprint spectrum are shown in the figure. Two chromatographic peaks were identified using the standard medicinal herb Daqingye as a control. By comparing with the control sample, the characteristic peaks were identified. It was found that peak 1 was hesperidin, and peak 2 was isoquercetin (Appendix A). The similarity results revealed high homology across different treatment time periods, with only slight differences in content, indicating that abiotic stress treatment impacts gene expression.

### 3.8. The Effect of Alkaline Stress Treatment on POD Activity and MDA Content in I. indigotica Tissue Samples

The *CCoAOMT* gene is associated with the synthesis of multiple methoxy flavones (PMFs), which, in turn, are related to antioxidant indicators. The activity of antioxidant enzymes, such as POD, is an essential indicator for evaluating the antioxidant level of plants. POD can clear excess reactive oxygen species and prevent oxidative damage. As shown in Figure 10, an extension of alkaline treatment times is associated with higher POD enzyme activity, which, in turn, strengthens the plant’s stress treatment resistance, indicating a significant relationship. MDA is a secondary product of membrane damage. When plants are subjected to stress treatment, MDA accumulates significantly at various time periods, and its content increases considerably at the cellular level. The above results indicate that the degree of membrane lipid peroxidation is high in 1d and 3d treatments, with a significant relationship, suggesting that growth conditions may influence it.

### 3.9. IiWrky48 and IiWrky54 Can Bind to the Promoter CCoAOMT Gene

Yeast one-hybrid is a typical method for detecting the interaction between proteins and DNA. To determine whether the promoter of the target gene can bind to *IiWRKY48* and *IiWRKY54*, a literature review revealed that the *Arabidopsis WRKY* gene (AT2G30250.1) regulates enzyme genes. Therefore, all *WRKY* genes of *Isatis indigotica* were co-constructed with it. As shown in Appendix A, *IiWRKY48* and *Arabidopsis WRKY* are on the same branch, and yeast one-hybrid experiments were conducted. The IiWRKY48-JG4-5 and IiWRKY54-JG4-5 constructs, which contain two promoters, and the pLacZi-IiCCoAOMT4 construct were co-transformed into the yeast strain EGY48, as shown in Figure 11. The yeast that was co-transformed with IiWRKY48-JG4-5, IiWRKY54-JG4-5, and pLacZi-IiCCoAOMT4 grew well, indicating that IiWRKY48-JG4-5 and IiWRKY54-JG4-5 can directly interact with the promoter of the pLacZi-IiCCoAOMT4 gene, and the colonies turned blue. However, the IiWRKY48-JG4-5 domain and IiWRKY54-JG4-5 cannot interact with other pLacZi-IiCCoAOMT domains. This result indicates that the IiWRKY48-JG4-5 and IiWRKY54-JG4-5 domains can directly interact with the domain of pLacZi-IiCCoAOMT4 and regulate its transcription.

## 4. Discussion

*CCoAOMT* is a class of S-adenosyl-L-methionine methyltransferases. The gene is crucial for plants’ physiological growth and their reaction to external stressors. The dried leaves (*Isatis indigotica Fort. folium*) and roots (*Isatidis radix*) of *I. indigotica* are traditional Chinese medicines, which have curative effects such as clearing heat and detoxification, cooling the blood, and resolving carbuncles. *I. indigotica* can also be used as an industrial dye. Therefore, *I. indigotica* is a medicinal and economic plant widely planted all over the world. Studying the regulatory mechanism of the growth and development of *I. indigotica* will help solve possible problems in *I. indigotica* breeding and improve the overall quality of *I. indigotica*.

### 4.1. Expansion of CCoAOMT Gene Family

The *IiCCoAOMT* gene family has not been identified yet, making the function of the *IiCCoAOMT* gene largely unclear. In this study, through bioinformatics techniques, a total of eight *IiCCoAOMT* genes were identified and classified into four subfamilies; notably, subgroup Ic contains no genes. The number and subgroup classification of these members are similar to those of other species. For example, there are *CCoAOMT* genes in Marchantia paleacea [49] and Populus [47], and they are all divided into four subfamilies. These results all indicate that the *CCoAOMT* gene family is conserved during the long evolutionary process. However, there is no information about *CCoAOMT* in *Isatis indigotica* [50,51]. Therefore, utilizing the *I. indigotica* genome document, we pinpointed the *IiCCoAOMT* gene family and examined the expression trends of an individual gene under alkali stress treatment.

### 4.2. Bioinformatics Analysis of the IiCCoAOMT Gene Family

In addition, *CCoAOMT* is vital in the production of lignin, flavonoids, and phenylpropanes. The conformation of a protein is determined by the gene’s structure, which, in turn, dictates its function, resulting in genes with analogous protein structures often performing analogous functions (Figure 2). According to the study and analysis, all *IiCCoAOMT* have one to eight structural motifs, which may be key to maintaining the functional preservation of the gene family. Except for *IiCCoAOMT2*, motif 5 exists in all other *IiCCoAOMTs*. Given the vital role of motif sequences at both ends in enzyme function and substrate attachment, the lack of motif five could result in the distinct functionality of *IiCCoAOMT2*. The protein structures of *IiCCoAOMT1*, *IiCCoAOMT6*, and *IiCCoAOMT7* are the same, and the exon–intron gene distribution is also consistent. *IiCCoAOMT3*, *IiCCoAOMT4*, and *IiCCoAOMT5* have specific institutional motifs, which may lead to the diversification of *IiCCoAOMT* gene functions. Gene duplication is considered to be the main factor leading to the expansion of gene families and the diversity of gene functions. The chromosomal examination of eight *IiCCoAOMT* gene locations revealed a dense distribution of three *IiCCoAOMT* genes on chromosome 7, indicative of proximal gene duplication, akin to *Arabidopsis thaliana* and poplar [52], and potentially linked to gene evolution. The remaining *IiCCoAOMT* genes are scattered on chromosomes 3 and 6 (Figure 5). Within poplar trees, the *CCoAOMT* genes are uniformly dispersed across chromosomes 1, 8, 9, and 10 [46].

Cis-regulatory components within the gene promoter area are crucial in controlling patterns of gene expression [53]. Research indicates that the Mybplant motif, located in the promoter region of genes responsible for phenylpropane biosynthesis, controls lignin production by attaching to the P-box element [54]. In our study, both *IiCCoAOMT4* and *IiCCoAOMT5* contain P-box elements. These genes are thought to participate in lignin synthesis. In addition, cis-acting elements related to ABA, GA, MeJA, and light responses were also found in the promoter of the *IiCCoAOMT* gene. Treatment with MeJA swiftly enhances the expression of lignin synthase genes, such as *CCoAOMT*, and fosters the synthesis of lignin [55,56]. Furthermore, ABRE and W-box have been verified as key cis-acting components in the promoter triggered by alkali stress treatment [42]. In most of the eight genes of *IiCCoAOMTs*, there are ABA response elements (ABREs). The W-box’s cis-acting element binding site under salt stress treatment suggests their possible involvement in *I. indigotica’s* reaction to alkali stress treatment. Consequently, we analyzed the *CCoAOMT* gene’s expression under salt stress treatment in *I. indigotica* and examined the potential regulatory mechanism.

Based on phylogenetic tree analysis, it has been shown that *AtCCoAOMT6* is involved in the biosynthesis of phenylpropanoid polyamine polymers in *Arabidopsis* [57]. Research has confirmed *AtCoAOMT7’s* involvement in the creation of phenylpropane and flavonoids, showing a marked inclination towards the methylation of both flavonoids and dihydroflavonoids [58]. Therefore, we speculate that *IiCCoAOMT1*, *IiCCoAOMT5*, *IiCCoAOMT6*, and *IiCCoAOMT7* are on the same branch as *AtCCoAOMT7* and may be involved in the biosynthesis of flavonoids. According to qRT-PCR, *IiCCoAOMT1* and *IiCCoAOMT8* are consistent with the expression trends of luteolin and isoquercetin, respectively. The *CCoAOMT* gene is also associated with the synthesis of multiple methoxyflavones (PMFs), which in turn are related to antioxidant indicators. The *CCoAOMT* gene plays a crucial role in plants, particularly in dicotyledonous plants, where it is involved in the synthesis of multiple methoxyflavones (PMFs). Multimethoxyflavones are a class of compounds with various biological activities, such as antioxidant, anti-inflammatory, and anticancer activities, which are of great significance to human health. It is speculated that the *CCoAOMT* gene regulates the production of multimethoxyflavonoids and has effects in antioxidant, anticancer, anti-obesity, and anti-inflammatory aspects. Consistent with the changes in *IiCCoAOMT3*, *IiCCoAOMT6*, and *IiCCoAOMT7* in qRT-PCR results, the trend in MDA oxidation index contents is consistent with that of *IiCCoAOMT2* contents. These genes may play a role in antioxidant activity.

### 4.3. Response of the IiCCoAOMT Family to Alkaline Treatment

When plants are subjected to alkaline stress treatment, the expression and activity of this gene undergo significant changes, becoming an essential part of plant defense mechanisms and maintaining internal balance. A high pH environment can damage the membrane structure and function of roots and interfere with ion absorption and balance, particularly leading to the fixation and loss of essential elements such as iron and phosphorus and thereby causing a series of physiological metabolic disorders. To cope with this stress treatment, plants activate complex signaling networks, among which the response of the *CCoAOMT* gene is particularly critical. It can rely on S-adenosylmethionine to catalyze the generation of various secondary metabolites, such as flavonoids, alkaloids, and phytohormones. Research has shown that O-methylation modification can enhance the affinity of flavonoids for lipids and proteins, expand the distribution range of flavonoids in cells, and be beneficial for improving the adaptability of plants to stress treatment during growth and development [59,60].

In our study, we detected the expression levels of the *CCoAOMT* gene in Isatis indigotica under alkaline stress treatment at the same concentration but different times: *IiCCoAOMT3*, *IiCCoAOMT4*, *IiCCoAOMT6*, and *IiCCoAOMT7*. The overall expression of these genes showed an upward trend, indicating that alkaline treatment increased the expression of these genes.

This study observed that the expression patterns of *IiCCoAOMT3* and *IiCCoAOMT8* were highly correlated with the accumulation trends of isoquercetin and luteolin, strongly suggesting their key roles in flavonoid B-ring methylation modification. This correspondence may stem from precise transcriptional regulation, where upstream transcription factors (*WRKY*) are activated under specific stress or developmental signals, thereby synergistically regulating the expression of *IiCCoAOMT* genes and the entire flavonoid pathway’s structural genes. At the same time, post-transcriptional regulation or post-translational modifications of enzyme activity may also fine-tune its final function. In addition, as a key node enzyme in the phenylpropane metabolism network, *CCoAOMT* simultaneously serves the synthesis of lignin monomers and some flavonoids. For example, many *CCOAOMT* genes in *VvCCoAOMT4* and other plant species [61] are involved in the synthesis of flavonoids, particularly anthocyanins. *CcWRKY* activates the transcription of *MYB* by binding to its promoter W-box, increasing the content of capsaicin and demonstrating its critical role in capsaicin synthesis [62]. Other research has shown that inhibiting *CcWRKY25* significantly decreases the transcription of *PAL* and *CBG* genes, which are crucial in the production of phenylpropane [63]. Therefore, upregulation of *IiCCoAOMT3/8* expression may trigger competitive allocation of common substrates (such as caffeoyl CoA) between lignin and flavonoid synthesis pathways. The significant increase in flavonoid content in this study may indicate that under specific physiological conditions, the enzyme is preferentially allocated to the flavonoid branch pathway, or its enzymatic properties are more inclined to catalyze flavonoid methylation. To fundamentally verify the physiological function of the *IiCCoAOMT* gene and further explore the above mechanisms, the core research direction will be the use of CRISPR/Cas9 technology to create functional deletion mutants or construct overexpression transgenic plants in the future. In the mutant, we expect to observe a significant reduction in methylated flavonoids and the accumulation of intermediate products such as luteolin. In overexpression or heterologous expression systems, we expect to see an increase in the content of target flavonoids.

## 5. Conclusions

In this study, we accurately located eight *IiCCoAOMT* genes in *I. indigotica*. We extensively investigated the genetic structure, domains, and preserved motifs of *CCoAOMT* to reveal its evolutionary connections across different plants. In addition, our study showed changes in gene expression patterns during exposure to alkaline stress treatment at various time intervals. Through yeast one-hybrid experiments, the *IiCCoAOMT* gene is likely involved in regulating the biosynthesis of flavonoids. Due to resource and time constraints, we mainly use Y1H for large-scale screening and qualitative identification, which is a limitation of our research. We plan to explore the details of these interactions further through quantitative enzyme activity measurement and mutation analyses of key sites in subsequent research. Despite this limitation, our Y1H data can support our main conclusion. In summary, this study provides a new perspective on the prospective characteristics of the *IiCCoAOMT* gene. In addition, this lays the foundation for a more comprehensive analysis of the stress response process of the *CCoAOMT* gene in *I. indigotica*.

## Figures and Tables

**Figure 1 biology-14-01518-f001:**
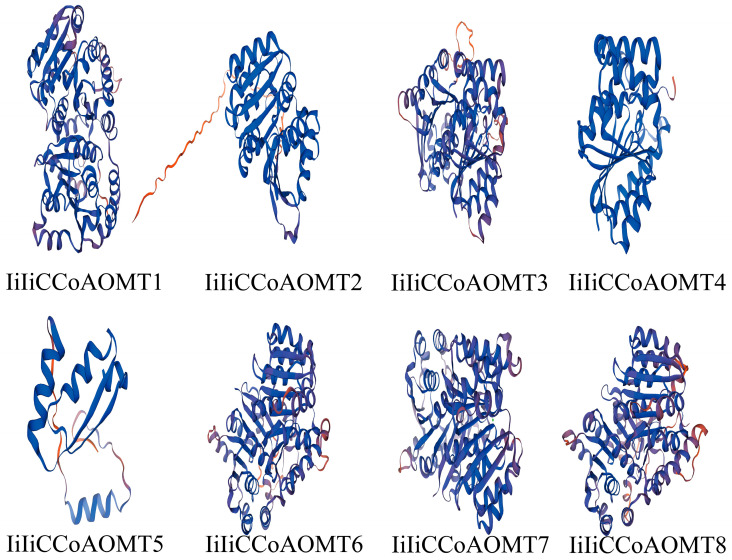
Tertiary structure analysis of IiCCoAOMT proteins.

**Figure 2 biology-14-01518-f002:**
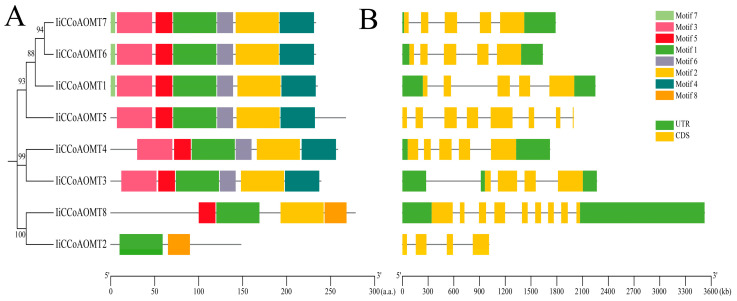
Structure of *IiCCoAOMT* gene and conserved motif analysis of its encoded protein. (**A**) *IiCCoAOMT* conservative domain analysis; (**B**) exon–intron structure analysis.

**Figure 3 biology-14-01518-f003:**
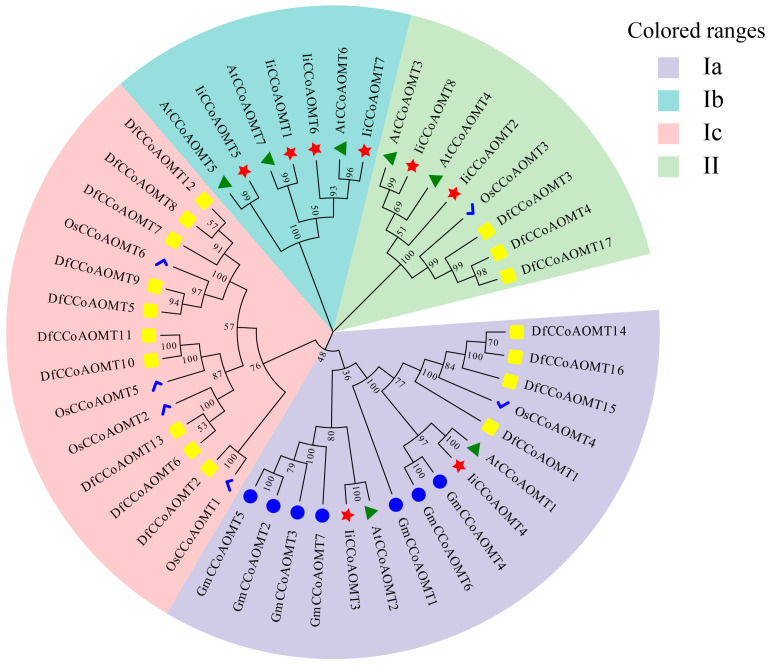
*CCoAOMTs* phylogenetic tree of *I. indigotica*, *Arabidopsis. thaliana*, *Dendrocalamus farinosus*, *Oryza sativa*, and *Glycine max*. In total, 45 CCoAOMT proteins were constructed using the neighbor-joining (NJ) method. The CCoAOMT protein is divided into Ia, Ib, Ic, and II. The proteins of I. indigotica are labeled with red, and various colors and shapes distinguish different varieties of *CCoAOMTs*.

**Figure 4 biology-14-01518-f004:**
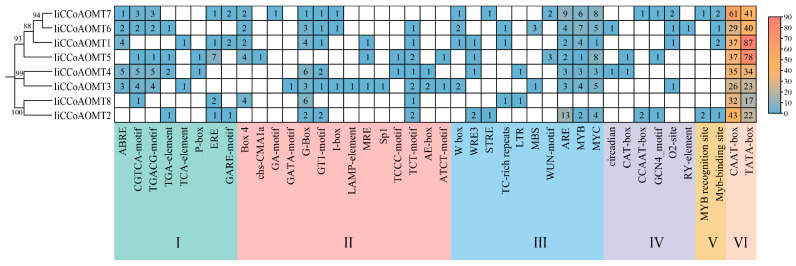
Analysis of cis-acting elements of the *IiCCoAOMT* gene family. Different colors in the IiCCoAOMTs promoter region indicate cis elements with various functions.

**Figure 5 biology-14-01518-f005:**
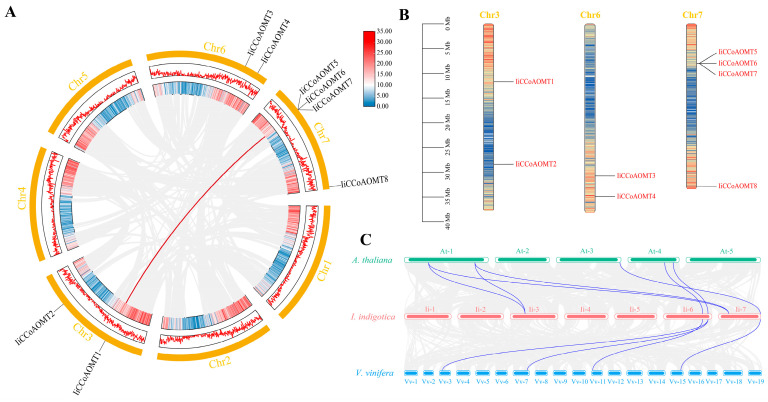
Chromosome localization and collinearity analysis of the *IiCCoAOMT* gene. (**A**) Intraspecific collinearity analysis of the *IiCCoAOMTs* gene. The gray background shows all the same linear segments in the genome of *I. indigotica*. The red line shows the identical linear *IiCCoAOMT* gene pairs. (**B**) The number of chromosomes of the *IiCCoAOMTs* gene is shown on the left side of each chromosome. The respective genes are marked on the right side of the chromosome. (**C**) Interspecific collinearity of *IiCCoAOMT*. Duplicate *IiCCoAOMT* gene pairs are linked with red lines.

**Figure 6 biology-14-01518-f006:**
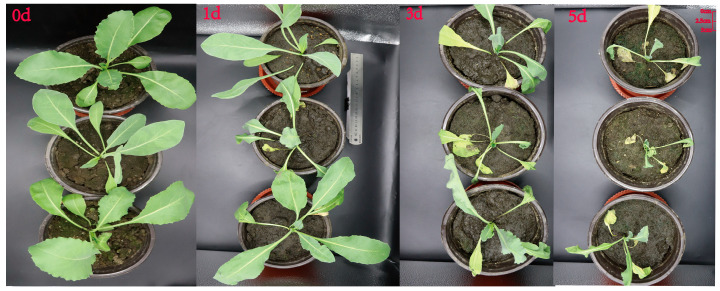
Phenotype of *I. indigotica* plants at different time periods under alkaline treatment. The phenotype of *Isatis indigotica*. Use a 200 mmol NaHCO_3_ alkaline solution with a pH value of 8.2 for treatment. With an increase in processing time, the plant height of *I. indica* decreased, the leaves wilted, and the growth rate slowed down.

**Figure 7 biology-14-01518-f007:**
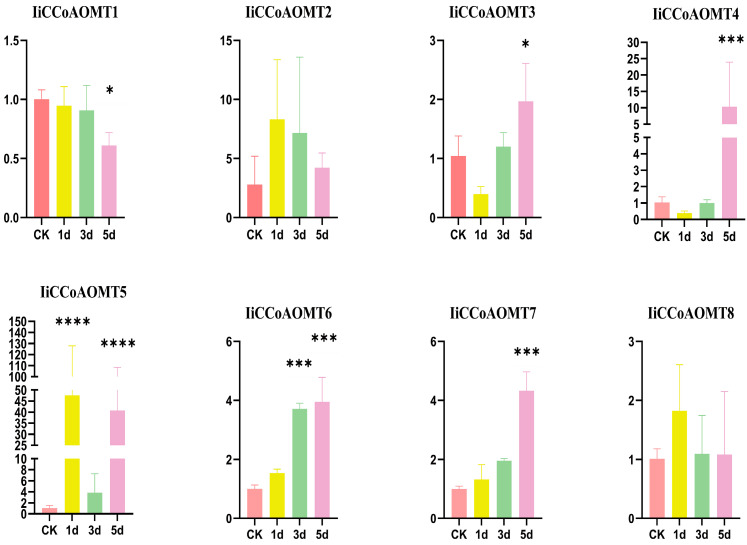
Quantitative RT-PCR analyses of *IiCCoAOMT* genes. Testing through significance analysis method. Quantitative RT-PCR results were normalized for *IiCCoAOMT* homeostasis. The data is the average of three replicates (mean ± SD, *n* = 3). The significance of chemical content in leaves of *I. indigotica* under alkaline stress treatment was studied using one-way analysis of variance and Tukey’s multiple-comparison test. (* *p* < 0.05; *** *p* < 0.001; **** *p* < 0.0001).

**Figure 8 biology-14-01518-f008:**
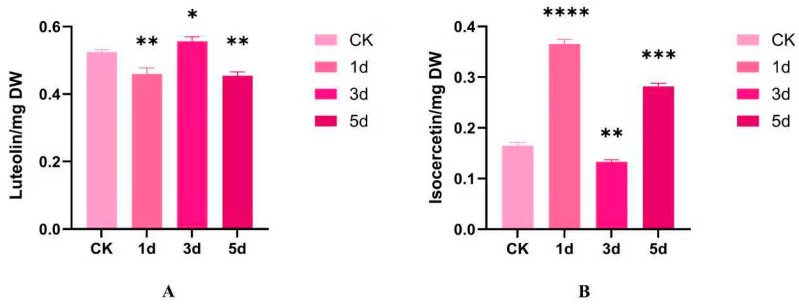
Analysis of changes in flavonoid content. (**A**) Luteolin content. (**B**) Isocercetin content. The data is the average of three replicates (mean ± SD, n = 3). The significance of chemical content in leaves of *I. indigotica* under alkaline stress treatment was studied using one-way analysis of variance and Tukey’s multiple-comparison test. (* *p* < 0.05; ** *p* < 0.01; *** *p* < 0.001; **** *p* < 0.0001).

**Figure 9 biology-14-01518-f009:**
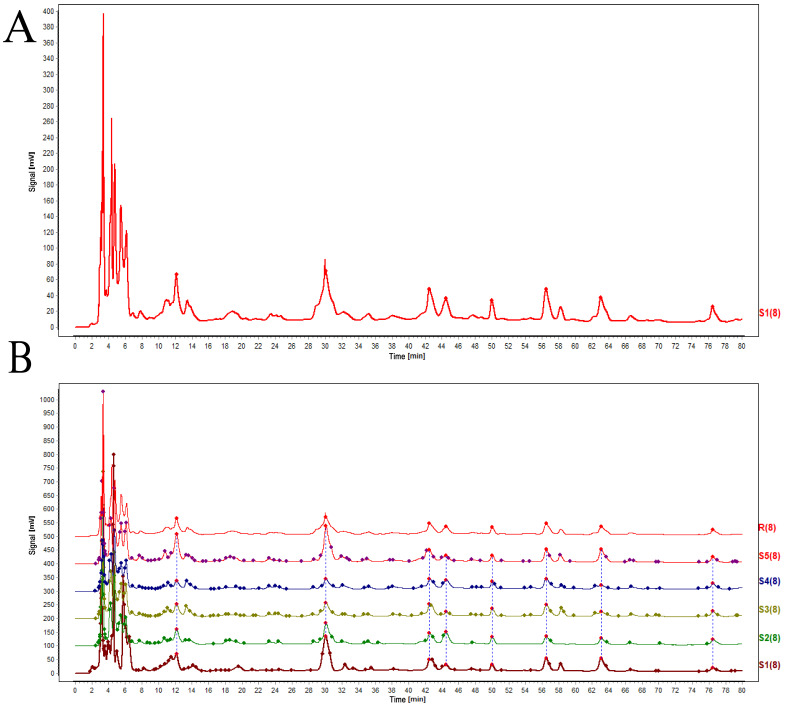
UPLC fingerprint spectrum of the *Isatis indigotica* sample. Use the ultrasonic extraction method to extract the sample, and transfer the extraction solution to a liquid phase bottle for measurement. (**A**) The peak spectrum of *I. indigotica* as a reference, and (**B**) analysis of the fingerprint spectra and reference drugs of *I. indigotica* at different stages.

**Figure 10 biology-14-01518-f010:**
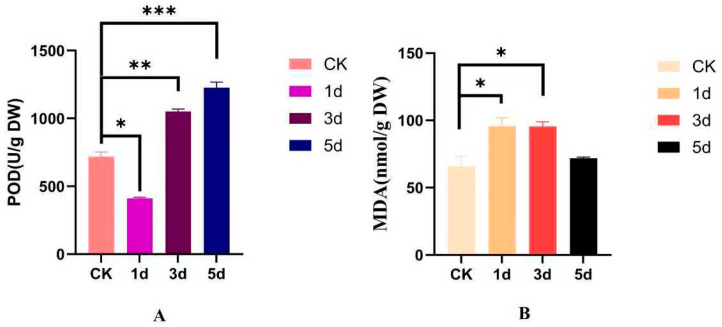
(**A**) POD activity; (**B**) MDA content. Data are means (mean ± SD, n = 3) of three replicates. Significance of SOD enzyme activity was analyzed using one-way ANOVA followed by Tukey’s multiple-comparisons test. (* *p* < 0.05; ** *p* < 0.01; *** *p* < 0.001).

**Figure 11 biology-14-01518-f011:**
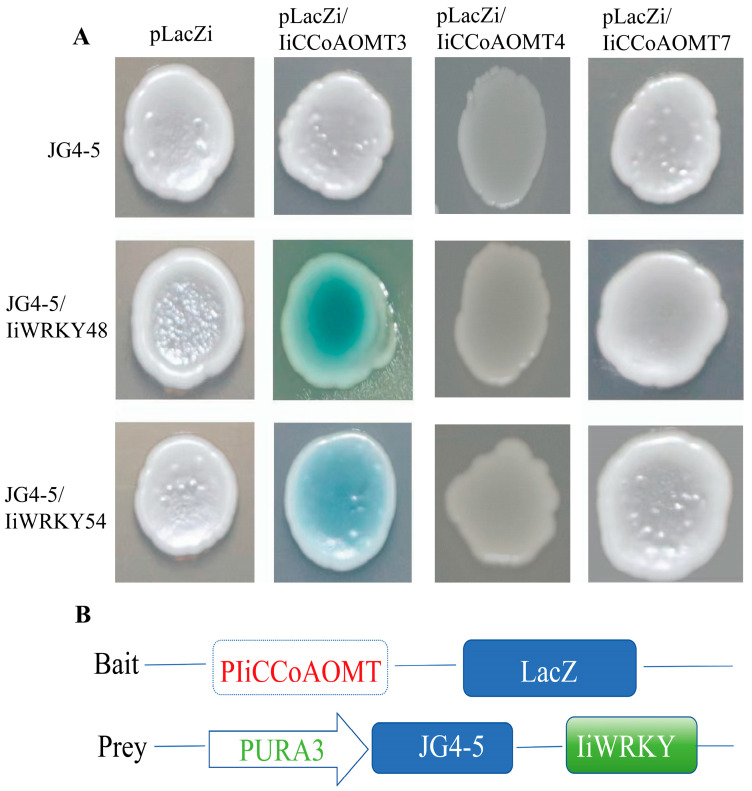
Yeast one-hybrid screening of target genes that bind to *IiWRKY48* and *IiWRKY54*. When pLacZi IiCCoAOMT3 co-transformed with pJG4-5-IiWRKY48 and pJG4-5-IiWRKY54, the colony appeared blue, indicating that transcription factors *IiWRKY48* and *IiWRKY54* interact with the *IiCCoAOMT3* gene. (**A**) shows the interaction between bait protein and prey protein in yeast one hybrid experiments. (**B**) shows a schematic diagram of the IiCCoAOMT and IiWRKY combination interaction.

**Table 1 biology-14-01518-t001:** The information of *IiCCoAOMT* members.

Gene Name	Accession Number	Chromosome	Number of Amino Acids	Protein Length (aa)	Molecular Weight (KDa)	Isoelectric Point	Subcellular Localization Predicted
IiCCoAOMT1	Iin23419.t1	Chr3	774	257	28.94	5.21	Specific location not predicted
IiCCoAOMT2	Iin27867.t1	Chr3	834	277	30.77	8.84	Chloroplast. Cytoplasm
IiCCoAOMT3	Iin25371.t1	Chr6	699	232	26.19	4.96	Chloroplast
IiCCoAOMT4	Iin25370.t1	Chr6	699	232	26.31	5.17	Chloroplast
IiCCoAOMT5	Iin11784.t1	Chr7	444	147	16.67	9.37	Nucleus
IiCCoAOMT6	Iin10582.t1	Chr7	705	234	26.45	4.97	Chloroplast
IiCCoAOMT7	Iin22645.t1	Chr7	717	238	26.70	5.27	Peroxisome
IiCCoAOMT8	Iin25369.t1	Chr7	801	266	30.24	5.48	Cytoplasm

## Data Availability

All data generated or analyzed in this study are included in the main text and its Appendix A.

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
