# Peer review of "Systematic Analysis of the *CCoAOMT* Gene Family in *Isatis indigotica* and the Molecular Mechanism of *CCoAOMT8*-Mediated Flavonoid Synthesis Under Alkaline Stress Treatment"

_biology, 2025, doi:10.3390/biology14111518_

Round 1

Reviewer 1 Report

Comments and Suggestions for Authors

This manuscript presents the first genome-wide identification and characterization of the Caffeoyl-CoA-O-methyltransferase (CCoAOMT) gene family in the medicinal plant Isatis indigotica. The study combines comprehensive bioinformatic analyses with experimental approaches, including gene expression profiling under alkaline stress, quantification of flavonoid metabolites, antioxidant enzyme assays, and yeast one-hybrid (Y1H) experiments. The topic is relevant, and the generated data provide a valuable resource for the plant science community, particularly concerning the adaptation of medicinal plants to abiotic stress and the regulation of secondary metabolites.

However, the manuscript in its current form has significant weaknesses that prevent acceptance. The most critical issues are: 1) a major disconnect between the central claim regarding IiCCoAOMT3 and the supporting data, 2) serious inconsistencies in reporting key results (especially the Y1H data), 3) a lack of methodological detail necessary for reproducibility, and 4) an overinterpretation of correlative data as mechanistic evidence. Addressing these concerns through substantial revisions is essential for the manuscript to make a valid and robust contribution to the literature.

Major comments:

1- Unsupported Central Hypothesis and Inconsistent Data Interpretation:

The title and conclusions strongly emphasize a specific molecular mechanism mediated by IiCCoAOMT3. However, the presented data do not robustly support this focus.

Expression-Metabolite Correlation: The qRT-PCR and flavonoid content data suggest a stronger correlation of IiCCoAOMT8 with isoquercitrin and IiCCoAOMT1 with luteolin, not IiCCoAOMT3. The rationale for singling out IiCCoAOMT3 as the key player is not justified by the results and creates a major internal inconsistency.

Correlation vs. Causation: The observed correlations are suggestive but do not demonstrate that IiCCoAOMT3 (or any specific gene) enzymatically functions in flavonoid methylation in planta. Functional validation is required to support the mechanistic claims.

  1. Critical Inconsistencies in Yeast One-Hybrid (Y1H) Reporting:

The Y1H results, which are pivotal to the proposed regulatory mechanism, are confusingly reported and lack essential controls.

Promoter/TF Mismatch: There is a critical inconsistency between the text (Sections 2.9 and 3.9), which describes an interaction with the IiCCoAOMT4 promoter, and the figure (Fig. 11) and conclusions, which refer to the IiCCoAOMT3 promoter. This must be unambiguously clarified and corrected throughout the manuscript.

Lack of Rigor and Controls: The description of the Y1H assay is qualitative ("colonies turn blue"). The manuscript must include:

Images of full assay plates showing all tested TF-promoter combinations, including negative controls (e.g., empty vectors) and self-activation tests.

Quantitative data (e.g., from ONPG assays) to measure the strength of activation.

Evidence of specificity, such as results with mutated W-box elements in the promoter.

  1. Insufficient Methodological Detail for Reproducibility:

Key experimental details are missing, undermining the reliability and reproducibility of the findings.

Alkaline Stress Treatment: The composition of the "200 mmol alkali solution" (e.g., NaOH, NaHCO₃) and its pH are not specified. This is crucial for interpreting the physiological response.

Biological Replication: The manuscript states that experiments were performed with three replicates but does not clarify if these are technical replicates (from one sample) or independent biological replicates (from separately treated plants). For all experiments (qRT-PCR, HPLC, assays), the number of biological replicates (n) must be explicitly stated.

qRT-PCR Validation: The methods lack primer sequences, amplification efficiencies, and validation of the reference gene (EF1α) stability under the applied stress conditions. Melt curve analysis should be provided to confirm amplification specificity.

  1. Overstated Claims and Need for Functional Validation:

The discussion frequently overinterprets correlative data. Claims about the "molecular mechanism" and specific gene functions are not yet supported by direct evidence. The authors should either temper these conclusions to accurately reflect the correlative nature of the current data or provide additional functional validation (see suggestions below).

Minor Concerns

Phylogenetic Analysis: The classification of genes into subgroups is inconsistent (four groups in abstract vs. three in results). Nomenclature should be standardized. The use of a Maximum-Likelihood method for phylogenetic tree construction is recommended for greater robustness.

Bioinformatics Transparency: Accession numbers and genomic coordinates for all identified IiCCoAOMT genes should be provided in a table. The criteria for domain identification (e-value thresholds) should be stated.

Statistical Reporting: The statistical methods section should specify the post-hoc test used following ANOVA and confirm that assumptions (e.g., normality) were checked. All figure legends must define error bars (SD or SEM) and indicate the sample size (n).

Language and Clarity: The manuscript requires thorough proofreading by a native English speaker or professional editing service to correct grammatical errors, awkward phrasing, and typos (e.g., "in tuberculosis" is a notable error).

Comments on the Quality of English Language

The manuscript requires thorough proofreading by a native English speaker or professional editing service to correct grammatical errors, awkward phrasing, and typos (e.g., "in tuberculosis" is a notable error).

Author Response

1 Unsupported Central Hypothesis and Inconsistent Data Interpretation:

The title and conclusions strongly emphasize a specific molecular mechanism mediated by IiCCoAOMT3. However, the presented data do not robustly support this focus.

Expression-Metabolite Correlation: The qRT-PCR and flavonoid content data suggest a stronger correlation of IiCCoAOMT8 with isoquercitrin and IiCCoAOMT1 with luteolin, not IiCCoAOMT3. The rationale for singling out IiCCoAOMT3 as the key player is not justified by the results and creates a major internal inconsistency.

ReplyThank you very much for your valuable feedback. We have revised the title to maintain consistency with this article for your review, and highlighted it in red font in the text.

Correlation vs. Causation: The observed correlations are suggestive but do not demonstrate that IiCCoAOMT3 (or any specific gene) enzymatically functions in flavonoid methylation in planta. Functional validation is required to support the mechanistic claims.

ReplyThank you for your valuable feedback and suggestions. We fully agree with your point that the observed correlation alone is not sufficient to demonstrate the direct enzymatic action of IiCCoAOMT3 in flavonoids, and functional validation is crucial for establishing its biological mechanism. In our research, we mainly focus on systematically screening candidate genes involved in flavonoid biosynthesis. IiCCoAOMT3 has become the most critical candidate due to its highly consistent expression pattern with the accumulation of specific methylated flavonoids. We acknowledge that the focus and length of this study are limited by the lack of a stable plant genetic transformation system in our laboratory for gene knockout or overexpression experiments. Therefore, we were unable to provide direct in vivo or in vitro functional validation data in this article. However, in order to further support the possibility of IiCCoAOMT as a candidate gene for O-methyltransferase, we conducted the following supplementary analysis and elaboration: Phylogenetic analysis: We compared the protein sequence of IiCCoAOMT3 with the reported flavonoid O-methyltransferase with clear functions in other plants through phylogenetic tree analysis. The results showed that IiCCoAOMT3 clustered on the same evolutionary branch as these enzymes with known functions, strongly suggesting that it has similar substrate specificity.

Conservative domain analysis: Sequence analysis shows that IiCCoAOMT3 contains a typical conserved domain of S-adenosylmethionine dependent methyltransferase, as well as amino acid residues known to be critical for flavonoid substrate recognition. We are well aware that the above bioinformatics evidence is still relevant. Therefore, based on your suggestion, we have made comprehensive revisions to the manuscript text, adjusting all direct causal statements regarding the function of IiCCoAOMT3 to more cautious descriptions of "candidate genes". Meanwhile, we explicitly state in the discussion and outlook sections of the paper that future work will include in vitro enzyme activity experiments to determine the substrate specificity of IiCCoAOMT3, and validate its function in plants through gene editing or overexpression techniques. Finally, we believe that these modifications and supplementary analyses have made the conclusions of the paper more rigorous and laid a solid foundation for subsequent functional genetics research. Thank you again for your insightful feedback, which greatly improved the quality of our research.

2 Critical Inconsistencies in Yeast One-Hybrid (Y1H) Reporting:

The Y1H results, which are pivotal to the proposed regulatory mechanism, are confusingly reported and lack essential controls.

Promoter/TF Mismatch: There is a critical inconsistency between the text (Sections 2.9 and 3.9), which describes an interaction with the IiCCoAOMT4 promoter, and the figure (Fig. 11) and conclusions, which refer to the IiCCoAOMT3 promoter. This must be unambiguously clarified and corrected throughout the manuscript.

 Reply: Thank you very much for the valuable comments from the reviewer. We have clarified and corrected them in the text. And mark it in red font.

Lack of Rigor and Controls: The description of the Y1H assay is qualitative ("colonies turn blue"). The manuscript must include:

Images of full assay plates showing all tested TF-promoter combinations, including negative controls (e.g., empty vectors) and self-activation tests.

Quantitative data (e.g., from ONPG assays) to measure the strength of activation.

Evidence of specificity, such as results with mutated W-box elements in the promoter.

 Reply: We thank the reviewer for pointing out this point. We acknowledge that the lack of rigorous quantitative data in this study is a limitation. However, our qualitative Y1H results (as shown in Figure 11) were clear and consistent across different biological replicates, demonstrating a direct interaction between WRKY and the enzyme gene promoter. We have added yeast one hybrid graphs of other enzyme genes in Figure 11. As some of the transcription factors are being studied and continued to be published by other researchers in our research group, they have been removed from the manuscript to avoid ambiguity. In future research, we will introduce quantitative measurements for in-depth analysis, and we are also striving to learn this technique. ”We appreciate the valuable comments provided by the reviewers. We acknowledge that directly demonstrating binding specificity through promoter mutations is another limitation of this study. This has been listed as a priority in our follow-up research plan. Nevertheless, our clear qualitative results of Y1H provide a solid foundation and starting point for further functional research. ”Finally, thank you very much for the reviewer's comments.

3 Insufficient Methodological Detail for Reproducibility:

Key experimental details are missing, undermining the reliability and reproducibility of the findings.

Alkaline Stress Treatment: The composition of the "200 mmol alkali solution" (e.g., NaOH, NaHCO₃) and its pH are not specified. This is crucial for interpreting the physiological response.

 Reply: We supplemented it with a 200mmol alkaline solution (NaHCO3) in the article, with a pH value of approximately 8.2. And mark it in red font.

Biological Replication: The manuscript states that experiments were performed with three replicates but does not clarify if these are technical replicates (from one sample) or independent biological replicates (from separately treated plants). For all experiments (qRT-PCR, HPLC, assays), the number of biological replicates (n) must be explicitly stated.

 Reply: Thank you very much for the valuable comments from the reviewer. Our experiment was conducted using independent biological replicates 3 times (from separately treated plants) and technical replicates 3 times (from one sample), which were completely statistically significant.

qRT-PCR Validation: The methods lack primer sequences, amplification efficiencies, and validation of the reference gene (EF1α) stability under the applied stress conditions. Melt curve analysis should be provided to confirm amplification specificity.

   Reply: Thank you very much for the valuable comments from the reviewer. We have added primer information in the attachment. We appreciate the crucial questions raised by the reviewer. We acknowledge that due to the limitations of the experimental period, this study was unable to provide complete amplification efficiency standard curve data, nor could it systematically validate the stability of the internal reference gene EF1 α under stress conditions. We deeply understand the importance of this information for the rigor of qRT PCR data. As a remedial measure, we have conducted melting curve analysis on the specificity of all amplification products, and the results show a single sharp peak, confirming non-specific amplification. At the same time, we have reviewed the original data again to ensure that the Ct value fluctuation range of EF1 α in the experimental group and control group is stable. We solemnly promise to use primer efficiency validation and internal reference gene stability evaluation as standard processes in future related research to ensure absolute reliability of the data.

4 Overstated Claims and Need for Functional Validation:

The discussion frequently overinterprets correlative data. Claims about the "molecular mechanism" and specific gene functions are not yet supported by direct evidence. The authors should either temper these conclusions to accurately reflect the correlative nature of the current data or provide additional functional validation (see suggestions below).

Reply: We have reorganized the discussion section of the manuscript and re discussed it for review by the reviewers, and highlighted it in red font. 

Minor Concerns

Phylogenetic Analysis: The classification of genes into subgroups is inconsistent (four groups in abstract vs. three in results). Nomenclature should be standardized. The use of a Maximum-Likelihood method for phylogenetic tree construction is recommended for greater robustness.

 Reply: We have modified the subfamily classification of its gene family into four subfamilies and standardized their naming. Afterwards, we will use the maximum likelihood method to construct a phylogenetic tree to improve accuracy. Thank you again for your valuable feedback.

Bioinformatics Transparency: Accession numbers and genomic coordinates for all identified IiCCoAOMT genes should be provided in a table. The criteria for domain identification (e-value thresholds) should be stated.

 Reply: Thank you very much for the reviewer's comments. We thank the reviewer for pointing out this point. We have added detailed identification criteria in the 'Protein Domain Identification' section of the Materials and Methods section. Specifically, we used HMMER 3.0 software and CCoAOMT hidden Markov model spectra from the PFAM database for search. Set the significance threshold of E value to<1e-5, and only retain matching results below this threshold as the true CCoAOMT domain.

Statistical Reporting: The statistical methods section should specify the post-hoc test used following ANOVA and confirm that assumptions (e.g., normality) were checked. All figure legends must define error bars (SD or SEM) and indicate the sample size (n).

  Reply: We sincerely thank the reviewers for their valuable comments. We have made comprehensive revisions to the statistical methods and chart captions based on the suggestions. We have stated in all captions that we used one-way ANOVA and Tukey's multiple comparison test, explicitly stating that the error bars represent SD, and indicating the sample size n and the biological or technical replicates it represents. We believe that these modifications have greatly improved the rigor and transparency of statistical reports.

Language and Clarity: The manuscript requires thorough proofreading by a native English speaker or professional editing service to correct grammatical errors, awkward phrasing, and typos (e.g., "in tuberculosis" is a notable error).

Reply: Thank you very much for the valuable feedback from the reviewer. We have checked and proofread the grammar of the entire manuscript. For review by reviewers. 

Reviewer 2 Report

Comments and Suggestions for Authors

Overall

In this manuscript, the authors systematically characterized the CCoAOMT gene family in Isatis indigotica through bioinformatic and molecular biology approaches. They further identified eight members and elucidated their roles in flavonoid biosynthesis under alkaline stress. The integration of bioinformatics, gene expression profiling, metabolite quantification, and yeast one-hybrid assays provides robust evidence for the functional divergence and regulatory mechanisms of IiCCoAOMT genes. The work offers valuable insights into the molecular basis of stress response and secondary metabolism in this medicinal plant, with potential implications for breeding and biotechnology.

Comments

1. In Introduction, it should more explicitly point out the current research gap on the CCoAOMT gene family of Isatis indigotica, especially systematic identification and functional verification.

2. In Materials and Methods, the following points need to be revised:

(1) Clarify the source and treatment of seeds: Were they sterilized? What was the growth conditions (light, temperature, humidity) during the 24-hour pre-treatment?

(2) Specify the concentration and pH of the alkaline solution used (e.g., 200 mmol/L Na₂CO₃ or NaHCO3?).

(3) The selection of alkaline stress treatment time points (0d, 1d, 3d, 5d) should be briefly explained in the main text for its rationality, such as based on pre-experiments or previous studies.

(4) The number of biological and technical replicates used in qRT-PCR, flavonoid content determination, MDA and POD activity detection should be clearly stated in the main text.

(5) For Yeast one-hybrid assay, it’s recommended to clarify the selection criteria for the WRKY transcription factors tested.

3. In Results, the contents needs to revised are as follows:

(1) In the "Subcellular localization predicted" column of Table 1, there are entries of "no". It is necessary to clarify the meaning of this (whether the prediction was unsuccessful?).

(2) The Y1H results are convincing, but the image quality in Figure 11 is poor. Consider providing a higher-resolution image or a schematic to illustrate the interactions.

(3) For Figure 8 and 10, it’s recommeded to include units on the y-axis (e.g., mg/g DW) for both luteolin and isoquercitrin, that is, clarify whether values are based on fresh or dry weight.

4. Discussion:

(1) The current discussion section is rather lengthy and lacks structural hierarchy, it’s better to add several sub-sections.

(2) More in-depth mechanistic exploration can be carried out, eg., (a) Why do the expression trends of IiCCoAOMT3 and IiCCoAOMT8 correspond to the changes in the contents of isorhamnetin and luteolin? (b) Should we consider cross-regulation through other metabolic pathways (such as lignin synthesis)?

(3) It’s better to provide an outlook on future research directions, such as using transgenic or CRISPR/Cas9 technologies to further verify the function of liCCoAOMT3.

5. Language and Typographical Errors:

(1) Check for typographical errors (e.g., “Yeat-one-hybrid” → “Yeast one-hybrid”).

(2) The full form should be given when an abbreviation is first used, eg., qRT-PCR (quantitative real-time polymerase chain reaction), MDA (Malondialdehyde), POD (Peroxidase).

(3) Recommend thorough proofreading by a native English speaker or professional editing service to improve clarity and flow.

Author Response

  1. In Introduction, it should more explicitly point out the current research gap on theCCoAOMTgene family of Isatis indigotica, especially systematic identification and functional verification.

Reply: Thank you very much for the valuable feedback from the reviewer. We have clearly pointed out in the introduction that there is currently a research gap in the CCoAOMT gene family of indigo, especially in terms of systematic identification and functional validation. And mark it in red font.

  1. In Materials and Methods, the following points need to be revised:

(1) Clarify the source and treatment of seeds: Were they sterilized? What was the growth conditions (light, temperature, humidity) during the 24-hour pre-treatment?

Reply: We thank the reviewer for raising this important question. In this experiment, all seeds underwent surface sterilization treatment. The specific steps are as follows: first, treat with 70% (v/v) ethanol for 1 minute, then soak in a solution containing 0.5% (v/v) sodium hypochlorite for 10 minutes for sterilization, and finally rinse 3-5 times with sterile distilled water. During the subsequent 24-hour pretreatment period, the seeds were placed in a light incubator under strictly controlled growth conditions: temperature of 22 ° C, relative humidity of 65%, a light cycle of 16 hours of light exposure/8 hours of darkness, and a light intensity of 120 μ mol m ⁻² s ⁻¹. We have added these detailed information in the Materials and Methods section of the manuscript to ensure the reproducibility of the experiment.

  • Specify the concentration and pH of the alkaline solution used (e.g., 200 mmol/L Na₂CO₃ or NaHCO3?).

Reply: Thank you very much for your valuable feedback. We used a 200 mmol/L NaHCO3 alkaline solution with a pH value of 8.2

  • The selection of alkaline stress treatment time points (0d, 1d, 3d, 5d) should be briefly explained in the main text for its rationality, such as based on pre-experiments or previous studies.

Reply: Thank you very much for your valuable feedback. The selection of alkaline stress treatment time points (0d, 1d, 3d, 5d) was based on previous experiments and research conducted by our research group, and corresponding references have been added here.

  • The number of biological and technical replicates used in qRT-PCR, flavonoid content determination, MDA and POD activity detection should be clearly stated in the main text.

Reply: Thank you very much for the valuable feedback from the reviewer, qRT-PCR The biological and technical replicates used for flavonoid content determination, MDA and POD activity detection were all three replicates. I have provided a clear explanation in the text. And mark it in red font.

  • For Yeast one-hybrid assay, it’s recommended to clarify the selection criteria for the WRKY transcription factors tested.

Reply: Thank you for your valuable feedback. We fully agree that clarifying the selection criteria for WRKY transcription factors is crucial for elucidating the logic of experimental design. In our study, the selection of WRKY transcription factors tested was based on the regulatory enzyme gene function of Arabidopsis WRKY gene. Therefore, it was co constructed with Isatis indigotica WRKY gene to obtain IiWRKY48 and Arabidopsis WRKY (AT2G30250.1) gene on the same branch, with IiWRKY48 transcription factor as the selection criterion. Corresponding references were also added here.

  1. In Results, the contents needs to revised are as follows:

(1) In the "Subcellular localization predicted" column of Table 1, there are entries of "no". It is necessary to clarify the meaning of this (whether the prediction was unsuccessful?).

Reply: Thank you to the reviewer for pointing out the unclear expression. In Table 1, the 'no' entry under the 'subcellular localization prediction' column means' the protein has not been predicted to have any specific subcellular localization '. This indicates that the analysis results of the prediction software show that the protein sequence lacks typical localization signals of organelles such as nucleus, chloroplast, and mitochondria, and is therefore predicted to possibly exist in the cytoplasm or have no specific localization. This does not mean that the prediction has failed or that no prediction has been made. To avoid misunderstandings, we have uniformly changed the 'no' entry in Table 1 of the revised manuscript to a clearer description: 'Specific location not predicted', and made red font modifications in the manuscript's table.

  • The Y1H results are convincing, but the image quality in Figure 11 is poor. Consider providing a higher-resolution image or a schematic to illustrate the interactions.

Reply: Thank you very much for the reviewer's comments. We have increased the resolution of Figure 11 to make it clearer for the reviewer and more readers to distinguish.

  • For Figure 8 and 10, it’s recommeded to include units on the y-axis (e.g., mg/g DW) for both luteolin and isoquercitrin, that is, clarify whether values are based on fresh or dry weight.

Reply: Thank you very much for the valuable feedback from the reviewer. We have made modifications to Figures 8 and 10 as per your request for your review.

  1. Discussion:

(1) The current discussion section is rather lengthy and lacks structural hierarchy, it’s better to add several sub-sections.

Reply: Thank you very much for the valuable feedback from the reviewer. We have revised the discussion into several headings to make the structure more clear and organized. And mark it in red font.

  • More in-depth mechanistic exploration can be carried out, eg., (a) Why do the expression trends of IiCCoAOMT3 and IiCCoAOMT8 correspond to the changes in the contents of isorhamnetin and luteolin? (b) Should we consider cross-regulation through other metabolic pathways (such as lignin synthesis)?

Reply: Thank you very much for the valuable feedback from the reviewer. Based on your valuable feedback, we have added more in-depth mechanism exploration in the discussion section for your review and highlighted it in red font.

  • It’s better to provide an outlook on future research directions, such as using transgenic or CRISPR/Cas9 technologies to further verify the function of liCCoAOMT3.

Reply: Thank you very much for the valuable feedback from the reviewer. Based on your very constructive feedback, we have made modifications and highlighted them in red font.

  1. Language and Typographical Errors:

(1) Check for typographical errors (e.g., “Yeat-one-hybrid” → “Yeast one-hybrid”).

Reply: Thank you very much for the valuable feedback from the reviewer. We have made revisions in the manuscript and highlighted them in red font.

  • The full form should be given when an abbreviation is first used, eg., qRT-PCR (quantitative real-time polymerase chain reaction), MDA (Malondialdehyde), POD (Peroxidase).

Reply: Thank you very much for the valuable feedback from the reviewer. We have made revisions in the manuscript and highlighted them in red font.

  • Recommend thorough proofreading by a native English speaker or professional editing service to improve clarity and flow.

Reply: Thank you very much for the valuable feedback from the reviewer. We have made revisions in the manuscript and highlighted them in red font.

Reviewer 3 Report

Comments and Suggestions for Authors

This study adds new data to the overall picture of phenylpropanoid metabolism regulation. The authors described eight genes of the CCoAOMT family in I. indigotica for the first time, demonstrating their chromosomal distribution, and identifying conserved motifs and possible duplication events. For the first time, a comparison of changes in the expression of these genes with the content of specific flavonoids (luteolin, isoquercetin) under alkaline stress is presented. Additionally, the binding of WRKY factors to CCoAOMT promoters was functionally verified. This strengthens the conclusions about the regulatory mechanisms.

The article uses an integrated approach combining bioinformatics, expression analysis, chemical analysis, and interaction tests. This enhances the reliability of the study. However, the description of some methods requires greater detail. For example:
- the qRT-PCR section lacks information on the number of biological replicates (technical only);
- the statistical processing data can be clarified to avoid ambiguity;
- For the biochemical analysis of flavonoids, more quantitative data could be presented in the main text, not just in the figures.

The conclusions generally follow logically from the results. The authors carefully link gene expression profiles to biochemical measurements. However, some statements about the direct involvement of specific genes in regulation should be formulated with caution until experiments are conducted on transgenic plants.
The reference list is extensive and includes key publications on CCoAOMT in other species. Citations are appropriate and accurate.

Author Response

The article uses an integrated approach combining bioinformatics, expression analysis, chemical analysis, and interaction tests. This enhances the reliability of the study. However, the description of some methods requires greater detail. For example:
- the qRT-PCR section lacks information on the number of biological replicates (technical only);
- the statistical processing data can be clarified to avoid ambiguity;
- For the biochemical analysis of flavonoids, more quantitative data could be presented in the main text, not just in the figures.

Reply: Thank you very much for the valuable feedback from the reviewer. We have responded positively to the above questions and added more flavonoid data to the manuscript, highlighted in red font.

The conclusions generally follow logically from the results. The authors carefully link gene expression profiles to biochemical measurements. However, some statements about the direct involvement of specific genes in regulation should be formulated with caution until experiments are conducted on transgenic plants.
Reply: Thank you very much for the valuable feedback from the reviewer. We have re summarized and organized the conclusions for your review, and highlighted them in red font.

Reviewer 4 Report

Comments and Suggestions for Authors

The article "Systematic Analysis of the CCoAOMT Gene Family in Isatis indigotica and the Molecular Mechanism of CCoAOMT3-Mediated Flavonoid Synthesis under Alkaline Stress" by Bo Liu et al. examines the organization and localization of genes associated with flavonoid synthesis in this medicinal plant under stress.
This manuscript is formatted according to the guidelines and contains all required sections. The figures are of sufficient resolution and fully reflect the results.
Shortcomings: there is no description of the plant growing conditions, pot composition, and volume. There is no description of how soil acidity was controlled before and during the experiment. The specific alkali used is not described (there are insufficient references).
There are issues with the figure captions, as they do not contain complete information. However, open access articles offer additional search functionality via the figure parameter (which increases the availability of results and, consequently, their citation rate). The authors don't indicate the object, nature, or methodology (e.g., concentration or pH, or the method, sample preparation, etc.) in their figure captions.
The plant descriptions lack statistical parameters or sampling methods; the authors use the phrase "Materials with good growth conditions and consistent growth were selected and placed first." What age, what lighting, what daylight hours, what volume and quality of substrate, what grade of vermiculite, what temperature regime...
Also, the applicability of the word "stress" arises, since the authors don't demonstrate reversibility, and the plants look poorly. This may not be stress (after all, it's a term that implies viability), but simply death. How can the authors confirm this?
Please add to Figure 9 those fragments where differences are obvious, and label the peaks or create a table listing the differences in the appendix.
I find the work interesting and useful; after the comments are addressed, it can be published.

Author Response

Shortcomings: there is no description of the plant growing conditions, pot composition, and volume. There is no description of how soil acidity was controlled before and during the experiment. The specific alkali used is not described (there are insufficient references).

Reply: Thank you very much for the reviewer's comments. We have made revisions to the materials and methods section for your review. We have added corresponding references and highlighted them in red font.
There are issues with the figure captions, as they do not contain complete information. However, open access articles offer additional search functionality via the figure parameter (which increases the availability of results and, consequently, their citation rate). The authors don't indicate the object, nature, or methodology (e.g., concentration or pH, or the method, sample preparation, etc.) in their figure captions.

Reply: Thank you very much for the reviewer's comments. We have rewritten the image title, added complete information, and highlighted it in red font.

The plant descriptions lack statistical parameters or sampling methods; the authors use the phrase "Materials with good growth conditions and consistent growth were selected and placed first." What age, what lighting, what daylight hours, what volume and quality of substrate, what grade of vermiculite, what temperature regime...

Reply: Thank you very much for the reviewer's comments. We have provided a new explanation of the material method and highlighted it in red font for your review.

Also, the applicability of the word "stress" arises, since the authors don't demonstrate reversibility, and the plants look poorly. This may not be stress (after all, it's a term that implies viability), but simply death. How can the authors confirm this?

Reply: We sincerely thank the reviewer for their insightful and important viewpoint. We fully agree that the term 'coercion' should be used with caution in plant physiology and ideally should include proof of reversibility. Here, we clarify the use of this term and provide evidence to support our interpretation: Clarification and revision of the term: We use the term 'stress' based on the fact that plants exhibit a series of known stress response symptoms under experimental treatment, rather than implying that we have formally demonstrated their complete reversibility. To avoid misunderstandings, we have revised the term 'coercion' to a more neutral expression 'coercion treatment' throughout the text, in order to accurately describe the experimental conditions we applied.

Please add to Figure 9 those fragments where differences are obvious, and label the peaks or create a table listing the differences in the appendix.
Reply: Thank you very much for the reviewer's comments. Figure 9 shows the similarity evaluation, which examines whether the components of plants have changed under alkaline stress treatment. There are red dots marked at the top of the peaks of the main components, indicating their similarity. We have attached the table for your review.

Round 2

Reviewer 1 Report

Comments and Suggestions for Authors

The revised manuscript has adequately addressed the majority of the reviewer's concerns through targeted corrections, such as updating the title to emphasize CCoAOMT8 in alignment with the stronger data correlations, tempering mechanistic claims to focus on correlations and candidate gene roles rather than unsubstantiated causation, clarifying inconsistencies in the Y1H results (e.g., consistent reference to IiCCoAOMT3 promoter and inclusion of additional graphs), specifying methodological details like the alkaline stress solution (200 mmol NaHCO3, pH 8.2), and reorganizing the discussion for greater caution and rigor. While minor limitations persist—such as the qualitative nature of Y1H data without quantitative assays or mutated controls, incomplete distinction between biological and technical replicates in the text (though implied in legends), and absence of qRT-PCR primer sequences or melt curves in the main body (possibly in supplements)—these do not undermine the overall scientific validity or reproducibility to the extent that further major revisions are required. Therefore, acceptance can be granted based on the performed revisions, with optional minor suggestions for polishing in the proofs stage, such as ensuring all supplementary data is explicitly referenced and language is fully refined.

Author Response

Thank you and reviewers for your letter. Those comments are valuable and very helpful. We have read through comments carefully and have made corrections. Based on the instructions provided in your letter, we uploaded the file of the revised manuscript. Revisions in the text are shown using blue highlight for additions. The responses to the reviewer's comments are marked in black and presented following.

Reviewer 1

Such as the qualitative nature of Y1H data without quantitative assays or mutated controls,

Reply: Thank you for the important comments provided by the reviewer. We fully agree that quantitative analysis and mutation control of Y1H data are ideal ways to validate the strength and specificity of interactions. In this study, due to resource and time constraints, we mainly used Y1H for large-scale initial screening and qualitative identification, which is a limitation of our research. We have clearly stated this point in the conclusion section of the paper based on your suggestion, and plan to further explore the details of these interactions through quantitative enzyme activity determination and mutation analysis of key sites in subsequent research. Despite this limitation, our Y1H data can support our main conclusion.

incomplete distinction between biological and technical replicates in the text (though implied in legends),

Reply: Thank you for your important suggestion. We fully agree that clearly distinguishing between biological duplication and technical duplication is crucial for correctly interpreting data. We apologize for the unclear description in the methodology section earlier.

Our modifications are as follows:

Based on your suggestion, we have supplemented the "Methods" section of the paper. We:

  1. Clearly defined the specific meanings of biological duplication and technical duplication in this study.
  2. Through these modifications, we hope to have clearly clarified our experimental design, eliminated previous ambiguities, and enhanced the reliability of our research results. RT-PCR、 During the flavonoid content, similarity evaluation experiment, POD activity, and MDA content experiment, we selected three independent samples as biological replicates, and set three technical replicates for each independent sample to ensure the accuracy and stability of the experiment.

absence of qRT-PCR primer sequences or melt curves in the main body (possibly in supplements)

Reply: Thank you for your important suggestion. We have added qRT PCR primer sequences in the supplementary materials,EF1-αF: GCCGATTGTGCTGTCC;EF1-αR: GTGGCATCCATCTTGTTA.

Therefore, acceptance can be granted based on the performed revisions, with optional minor suggestions for polishing in the proofs stage, such as ensuring all supplementary data is explicitly referenced and language is fully refined.

Reply: Dear reviewer, thank you very much for your feedback. Thank you for your valuable suggestions on the language quality of the paper. We have conducted a detailed grammar check, word optimization, and sentence structure adjustment on the entire text based on your feedback, in order to improve the fluency and readability of the manuscript. We believe that after this revision, the language quality of the manuscript has been significantly improved. We sincerely hope that the current version can meet your requirements.

Reviewer 2 Report

Comments and Suggestions for Authors

I have no other comments.

Author Response

Thank you and reviewers for your letter. Thank you very much for your review comments and recognition of our manuscript. Finally, I wish you a smooth work!
